# Insights into the self-assembly and interaction of sars-cov-2 fusion peptides with biomimetic plasma membranes

Nisha Pawar [1], Andreas Santamaria [2,3], Brigida Romano[2], Krishna C. Batchu[2], Valerie Laux[2], Eduardo Guzman [3,4], Nathan. R. Zaccai [5], Alberto Alvarez-Fernandez [1] ✉ & Armando Maestro [1,6] ✉

First identified in late 2019, the COVID-19 pandemic, caused by the SARS-CoV-2 coronavirus, rapidly escalated into a global health crisis. SARS-CoV-2 is a single-stranded RNA virus encased in a lipid envelope that houses key structural proteins, including the Spike glycoprotein, which mediates viral entry into host cells. Within Spike, the S2 subunit, and particularly its fusion domain, plays a critical role in merging viral and host membranes. To explore how receptor-driven Spike clustering influences this process, we investigated the self-assembly of S2 fusion peptides (FPs) and their interactions with biomimetic plasma membrane (PM) models composed of phospholipids, sphingomyelin, and cholesterol. Atomic force microscopy, laser direct infrared spectroscopy, neutron reflectometry, and grazing-incidence X-ray diffraction reveal that FPs form supramolecular assemblies that exclude cholesterol-rich nanodomains, increase membrane fluidity, and disrupt raft-like order associated with ACE2 binding. The appearance of spiral FP fibers supports a loaded-spring mechanism for membrane remodeling, offering a model for cooperative peptide-driven fusion, highlighting opportunities for antiviral and nanobiotechnological applications.

Fusion peptides (FPs) from the SARS-CoV-2 Spike (S) protein are critical for the virus's entry into host cells[1,2]. These short peptide sequences, located within the Spike protein´s fusion domain, display characteristic features that facilitate their interaction with lipid bilayers, enabling the merging of viral and host cell membranes and ultimately driving viral infection[3–5]. Beyond their biological relevance in viral entry, SARS-CoV-2 FPs have also shown promise as building blocks for the fabrication of advanced functional nanomaterials[6]. Their ability to self-assemble into well-defined nanostructures at fluid interfaces offers opportunities for designing chiral nanomaterials with potential applications in medicine and nanotechnology.

Structurally, the S protein consists of two subunits, S1 and S2, each with distinct roles in viral infection[7]. The S1 subunit primarily facilitates viral attachment to the host cell by binding to its receptors. In contrast, the S2 subunit drives the subsequent membrane fusion process, allowing the viral genome to enter the host cell[8]. Within the S2 subunit, a critical region known as the fusion domain interacts directly with the host cell´s lipid bilayer[9], disrupting and connecting opposing membranes to merge the viral and host membranes[10]. Intermolecular interactions between specific sections of the fusion domain, called FPs, and the lipid bilayer are fundamental for regulating fusion[11]. By embedding into the lipid bilayer, FPs induce membrane

[1]Centro de Fisica de Materiales (CFM-MPC), CSIC-EHU, Donostia-San Sebastian, Spain. [2]Institut Laue-Langevin, Grenoble, France. [3]Departamento de Química Física, Facultad de Ciencias Químicas, Universidad Complutense de Madrid, Ciudad Universitaria s/n, Madrid, Spain. [4]Instituto Pluridisciplinar, Universidad Complutense de Madrid, Madrid, Spain. [5]Cambridge Institute for Medical Research, University of Cambridge, Cambridge, United Kingdom. [6]IKERBASQUE-Basque Foundation for Science, Bilbao, Spain. ✉e-mail: alberto.alvarez@ehu.eus; armando.maestro@ehu.eus

destabilization, creating an initial point of contact that promotes the merging of the viral and host cell membranes.

The expected characteristics of FPs, short, hydrophobic, and potentially containing canonical fusion tripeptides (such as YFG or FXG), along with a central proline residue, have led to the identification of several putative SARS-CoV-2 FPs. Specifically, these sequences include FP1 (residues 816-837), FP2 (835-856), and FP4 (885-909), with the latter commonly referred to as the internal FP. Moreover, these peptides are not only integral to the fusion process but have also been identified as targets for broadly neutralizing antibodies against all known human-infecting coronaviruses. These molecules are therefore promising candidates in efforts toward universal coronavirus vaccines[12,13].

Given this intrinsic ability to interact with and perturb lipid membranes, FPs serve as ideal candidates for studying peptide-lipid interactions. Exploring their behavior at membrane interfaces can enhance our understanding of viral entry mechanisms while inspiring the development of biomimetic nanomaterials. In earlier work, we revealed that different SARS-CoV-2 FPs perform distinct functions upon interacting with plasma membrane (PM) monolayer models[11]. However, a mechanistic understanding of how fusion peptides self-assemble, particularly under physiologically relevant lipid packing conditions, and modulate membrane structure remains to be established.

Here, to gain further insights into the viral fusion process, including the effect of host receptor-associated Spike clustering[14], we investigate the self-assembly of SARS-CoV-2 FPs incubated with lipid mixtures that mimic the outer leaflet of the eukaryotic PM, creating hybrid peptide-lipid Langmuir monolayers. These PM models are characterized by a high cholesterol content (50 mol%), and include sphingomyelin (SM), zwitterionic phosphatidylcholines (PC) and phosphatidylethanolamines (PE), together with minor fractions of anionic phosphatidylserines (PS) and phosphatidylinositols (PI)[15]. By compressing monolayers from fluid to condensed phases, we simulate membrane packing states relevant to viral entry, including confined zones where FP oligomerization is expected. While structural studies indicate that only ~20–40 Spike trimers are present per virion in total[16], even partial accumulation of these molecules at a contact interface could raise FP density sufficiently to promote cooperative interactions. Laser Direct Infrared (LDIR) spectroscopy was used to resolve the secondary structure adopted by the peptides upon membrane association, while neutron reflectometry (NR) enable quantitative determination of the PM composition and its modulation upon peptide binding. Grazing incidence X-ray diffraction (GIXD) and atomic force microscopy (AFM) elucidated the in-plane molecular organization and lipid packing within the PM monolayer, as well as the resulting surface morphology following FP binding. These techniques revealed different FP oligomeric structures at the PM interface, including rigid fibres and spiral architectures. Such assemblies should be regarded as mechanistic proxies that illustrate how locally elevated FP density, whether achieved physiologically through clustering or experimentally in vitro, can drive supramolecular organization[14].

By modulating both fluidity and topology, FP oligomerization at the PM emerges as a mechanism optimized for mechanochemical efficiency: converting binding free energy into membrane stress and remodeling with minimal energetic cost. These principles, e.g., mechanical energy accumulation and pressure-triggered lipid reorganization, suggest design rules for responsive peptide-lipid systems. Beyond advancing our understanding of viral fusion, these insights provide a framework for developing adaptive nanomaterials capable of responding to membrane tension or stress.

## Results
The direct self-assembly of three different peptides derived from the SARS-CoV-2 S protein fusion domain, FP1, FP2, and FP4 (Fig. 1a), was investigated in the presence of lipid Langmuir monolayers mimicking the outer leaflet of the eukaryotic PM (Fig. 1b). The general methodology used in this study is outlined in Fig. 1c. Lipid monolayers, prepared with and without incorporating FPs, were formed at the air/water interface using a Langmuir-Blodgett (LB) trough[17]. Uniaxial compression within the LB trough modulated peptide-lipid interactions, enabling precise tuning of lipid packing density and controlling peptide self-organization at the interface. These monolayers were concurrently characterized in situ with NR and GIXD to assess their structural organization. The monolayers were then transferred onto freshly cleaved mica surfaces for AFM analysis to evaluate topographical and physico-chemical properties. The peptides' secondary structure, in both the presence and absence of lipids, was further analyzed using LDIR spectroscopy.

### Impact of fusion peptides on lipid packing in the outer leaflet of the plasma membrane
The phase behavior of PM and hybrid PM-FPs monolayers at the air-water interface was studied using surface pressure ($\Pi$) – area per molecule ($A$) isotherms (Fig. 1d), combining a LB trough with low $q_z$ NR measurements (Supplementary Fig. 1). The lateral compressibility of the monolayer was determined from the slope of the $\Pi$-A isotherm. This monolayer's intrinsic property, commonly expressed as the compressional elastic modulus[18] defined as $C_s^{-1} = -A(\partial \Pi / \partial A)$, is shown in Fig. 1f.

At low surface pressures, the PM monolayer exhibits an intrinsically disordered, fluid phase known as the liquid-expanded (LE) phase. In this regime, the compressional elastic modulus remains relatively constant at $C_s^{-1} \approx 74 \pm 3$ mN/m. A discontinuity in the isotherm at $\Pi \approx 20$ mN/m indicates a transition to a condensed (C) phase, driven by tighter lipid packing and enhanced lateral interactions. In the C phase, the elastic modulus increases markedly with surface pressure following a trend of $C_s^{-1} \approx 6\Pi$, consistent with stronger lipid-lipid interactions and reduced molecular mobility as the monolayer becomes more ordered.

The incorporation of FPs significantly alters the $\Pi$-A isotherm of the PM monolayers (Fig. 1d). For all three FPs studies (FP1, FP2, and FP4), the isotherm shifts to higher A values, indicating their interaction with the PM. This expansion suggests that the FPs intercalate into the monolayer, disrupting lipid packing and increasing the molecular area (Fig. 1e). This behavior can be attributed to a combination of factors, including hydrophobic residues intercalating into lipid acyl chains, charged residues interacting with headgroups, and potential peptide aggregation or supramolecular assembly.

Compared with FP2, FP1 and FP4 exhibit stronger interactions with the PM, as evidenced by greater isotherm expansion. This trend is consistent with reported binding affinities of these FPs for PM model membranes[11]. FP1 interacts favorably with both lipid headgroups and acyl chains due to its overall negative charge at physiological pH, as well as its hydrophobic residues (e.g., phenylalanine, leucine). The positively charged FP4, due to the presence of lysine and arginine residues, associates strongly with anionic PS and PI lipid headgroups. By contrast, FP2's neutral net charge limits electrostatic interactions, and its cysteine residues may form inter-peptide disulfide bonds, restricting conformational flexibility. These features reduce FP2's ability to integrate into and perturb the PM, resulting in weaker isotherm expansion.

The compressional elasticity data (Fig. 1f) further highlight the disruptive effect of FPs. In their presence, PM monolayers remain in the LE phase across the full pressure range. This persistent fluidity arises from the peptides' interference with lipid packing, introducing steric hindrance and preventing tight packing. Additionally, the amphipathic nature of the FPs would allow for partial peptide insertion into the monolayer, as observed previously by NR[11], leading to increased membrane disorder and fluidity. The interplay between

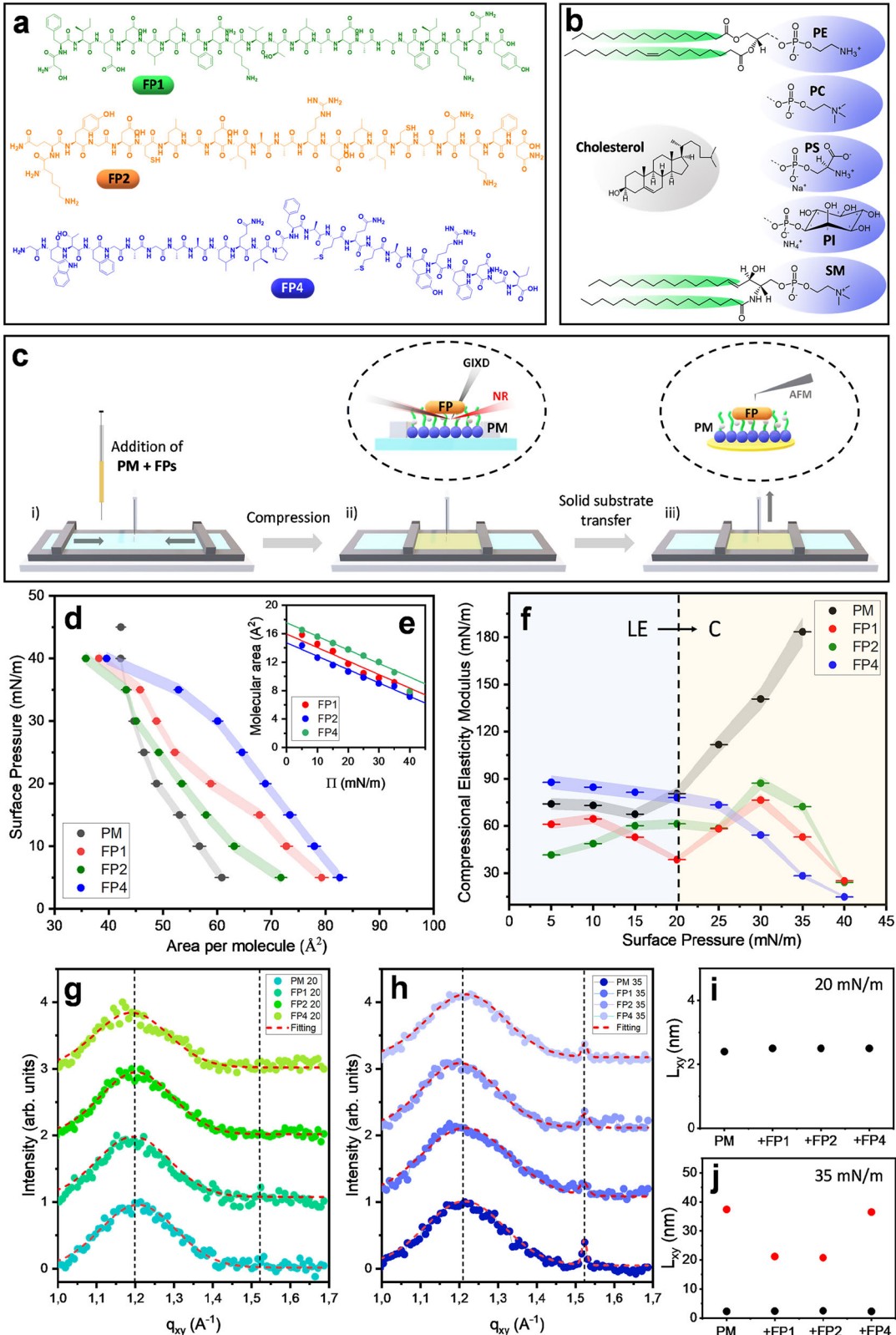

**Fig. 1 | Molecular components, assembly workflow, and biophysical characterization of Fusion Peptides - PM Monolayers.** Structure of the peptides (**a**) and lipids (**b**) used during this work. Schematic representation of the PM monolayer fabrication and the characterization techniques used during this study (**c**). Π-A compression isotherm [$n = 3$ different samples (mean ± SD)]. (**d**) FPs molecular area calculations (**e**), and compression modulus [$n = 3$ different samples (mean ± SD)].

(**f**) for the PM in the absence and presence of FP1, FP2, and FP4 spread at the air/water interface. Variation of diffracted GIXD intensity as a function of in-plane scattering vector component ($q_{xy}$) of PM, PM-FP1, PM-FP2, and PM-FP4 at a surface pressure of 20 mN/m (**g**) and 35 mN/m (**h**) and corresponding calculated $L_{xy}$ (**i**, **j**). Source data for all graphs included in this figure is provided as a Source Data file.

**Table 1 | Structural parameters obtained from the GIXD analysis for the samples studied during this work**

| $\Pi$ = 20 mN/m | | | | | |
|---|---|---|---|---|---|
| | $q_{xy} \pm 0.02$ (Å⁻¹) | $q_z \pm 0.02$ (Å⁻¹) | | Area ± 0.1 (Å²) | $L_{xy} \pm 0.1$ (nm) |
| PM | 1.21 | 0.05 | 6.01 | 31.3 | 2.4 |
| PM-FP1 | 1.20 | 0.05 | 6.07 | 31.9 | 2.5 |
| PM-FP2 | 1.20 | 0.05 | 6.03 | 31.5 | 2.5 |
| PM-FP4 | 1.19 | 0.06 | 6.09 | 32.2 | 2.5 |
| $\Pi$ = 35mN/m | | | | | |
| PM | 1.21 1.52 | 0.050 | 5.29 | 27.4 | 2.33 7.4 |
| PM-FP1 | 1.21 1.52 | 0.051 | 5.31 | 27.5 | 2.42 0.8 |
| PM-FP2 | 1.20 1.52 | 0.052 | 5.29 | 27.7 | 2.52 1.2 |
| PM-FP4 | 1.22 1.52 | 0.051 | 5.28 | 27.3 | 2.33 6.5 |

hydrophobic residues inserting into the lipid tails and hydrophilic residues interacting with the aqueous subphase creates a dynamic environment that inhibits the transition to a C phase. Despite their differences in charge and sequence, all three FPs consistently modulate the properties of the PM monolayer, underscoring their universal ability to disrupt lipid organization and maintain monolayer fluidity.

To probe in-plane lipid organization, GIXD measurements were performed on PM monolayers with and without FPs at $\Pi$ = 20 mN/m and 35 mN/m, corresponding to LE and C phases, respectively. Representative GIXD intensity profiles are shown in Fig. 1g, h, with full contour maps in Supplementary Figs. 2–3. At both surface pressures, a diffuse, broad Bragg peak centered at $q_{xy}$ = 1.20 ± 0.02 Å⁻¹ is observed (Fig. 1g, h), indicating limited short-range crystallinity within the monolayer. The corresponding in-plane correlation length ($L_{xy}$) slightly decreases from 24.5 ± 0.5 Å at 20 mN/m to 23.0 ± 0.5 Å at 35 mN/m (Fig. 1g, Table 1). This diffraction peak is characteristic of cholesterol-enriched domains in sterol-phospholipid mixtures, as the ones studied here[19]. The existence of this Bragg peak suggests that cholesterol molecules organize perpendicular to the interface in a hexagonal 2D lattice, consistent with prior GIXD studies[20,21]. Further support for this interpretation comes from the Bragg rod analysis, which a maximum near the horizon ($q_z \approx 0.05$ Å⁻¹, Supplementary Fig. 4), confirming the perpendicular orientation of cholesterol molecules relative to the air/water interface. The slight reduction in $L_{xy}$ with increasing $\Pi$ is attributed to compression-induced rearrangements of cholesterol molecules, reflecting their dominance in the lipid composition of the PM monolayer.

At $\Pi$ = 35 mN/m, a second, weaker diffraction peak appears at $q_{xy}$ = 1.52 ± 0.02 Å⁻¹, which likely corresponds to a hexagonal arrangement of lipid chains from the other lipids (PC, PE, PS, and possibly SM)[22,23]. The $L_{xy}$ for this lipid arrangement is 374 ± 1 Å, significantly larger than the $L_{xy}$ calculated for the cholesterol domains. This suggests that while cholesterol's rigid, bulky structure inherently limits long-range order, the more flexible acyl chains of phospholipids enable tighter packing and higher in-plane coherence. The coexistence of these two Bragg peaks reflects phase separation within the monolayer: cholesterol-rich regions coexist with more ordered domains of the other lipids. This separation is reinforced by compression, which promotes the organization of the phospholipid components while maintaining the short-range order typical of cholesterol. Such observations are consistent with prior reports on Langmuir monolayers of PE and cholesterol mixtures, where distinct crystalline domains were similarly detected[24].

Notably, the cholesterol-associated peak at $q_{xy} \approx 1.20$ Å⁻¹ remains unchanged in the presence of FP1, FP2, or FP4 (Fig. 1i, j), suggesting negligible direct interaction of the FPs with the ordered cholesterol domains. This aligns with theoretical studies indicating that the SARS-CoV-2 FPs preferentially associate with phospholipids over

cholesterol[25]. By contrast, the intensity of the phospholipid-associated peak at $q_{xy}$ = 1.52 Å⁻¹ is markedly weakened by FP1 and FP2, but less affected by FP4 (Fig. 1h). These results indicate that FP1 and FP2 disrupt acyl chain ordering, whereas FP4 primarily engages headgroups, allowing domain structure to be maintained.

## AFM analysis of the phase behavior and lipid organization in PM monolayers under compression

To investigate the in-plane lipid organization of PM monolayers in the absence of FPs, AFM was conducted using LB films as PM models. These monolayers were prepared at selected $\Pi$ of 20 mN/m and 35 mN/m, which correspond to the LE and C phases of the outer leaflet of the PM membrane, respectively. AFM micrographs of the pristine PM LB monolayer transferred onto freshly cleaved mica substrates at these surface pressures are presented in Fig. 2a, c.

The AFM topographical micrograph of the pristine PM monolayer at $\Pi$ = 20 mN/m (Fig. 2a) confirms its structural integrity and reveals a surface roughness of approximately 1 nm (Fig. 3b). This roughness indicates the presence of two different regions within the monolayer, suggesting lateral phase separation. These regions correspond to the coexistence of two liquid phases: the liquid-ordered (Lo) phase and the liquid-disordered (Ld) phase[26]. The Lo phase is characterized by high lipid packing, primarily driven by strong cohesive van der Waals interactions between the hydrophobic regions of cholesterol and SM. These interactions create tightly packed, less fluid domains enriched in cholesterol and SM, which appear as brighter areas in the AFM micrograph. In contrast, the Ld phase, represented by darker regions, is more disordered and fluid, with a composition predominantly consisting of PE, PC and PS, resulting in a lower cholesterol content. The differences in lipid composition and molecular interactions contribute to the lateral heterogeneity observed in the monolayer[11,24,27].

At $\Pi$ = 35 mN/m, compression of the PM monolayer reduces the available molecular area, limiting lipid mobility and inducing structural rearrangements consistent with the condensed phase[11]. AFM at this lateral pressure (Fig. 2c) and corresponding topographical profile (Fig. 2d) reveal that the bright, Lo nanodomains become smaller, more densely packed, and distributed more randomly across the monolayer. The increased concentration and irregular spatial arrangement of Lo domains likely result from a balance between domain coalescence, which is energetically favorable, and the spatial constraints imposed by compression. These constraints inhibit domain growth, leading to a fragmented organization.

This phase behavior can be explained by the thermodynamics of lipid packing under compression. At higher $\Pi$, the system reduced its free energy by increasing molecular packing density, stabilizing the Lo domains. However, the compression limits the available area for lipid molecules, favoring smaller, densely packed Lo domains rather than extended ones. The randomization of Lo domains' distribution under

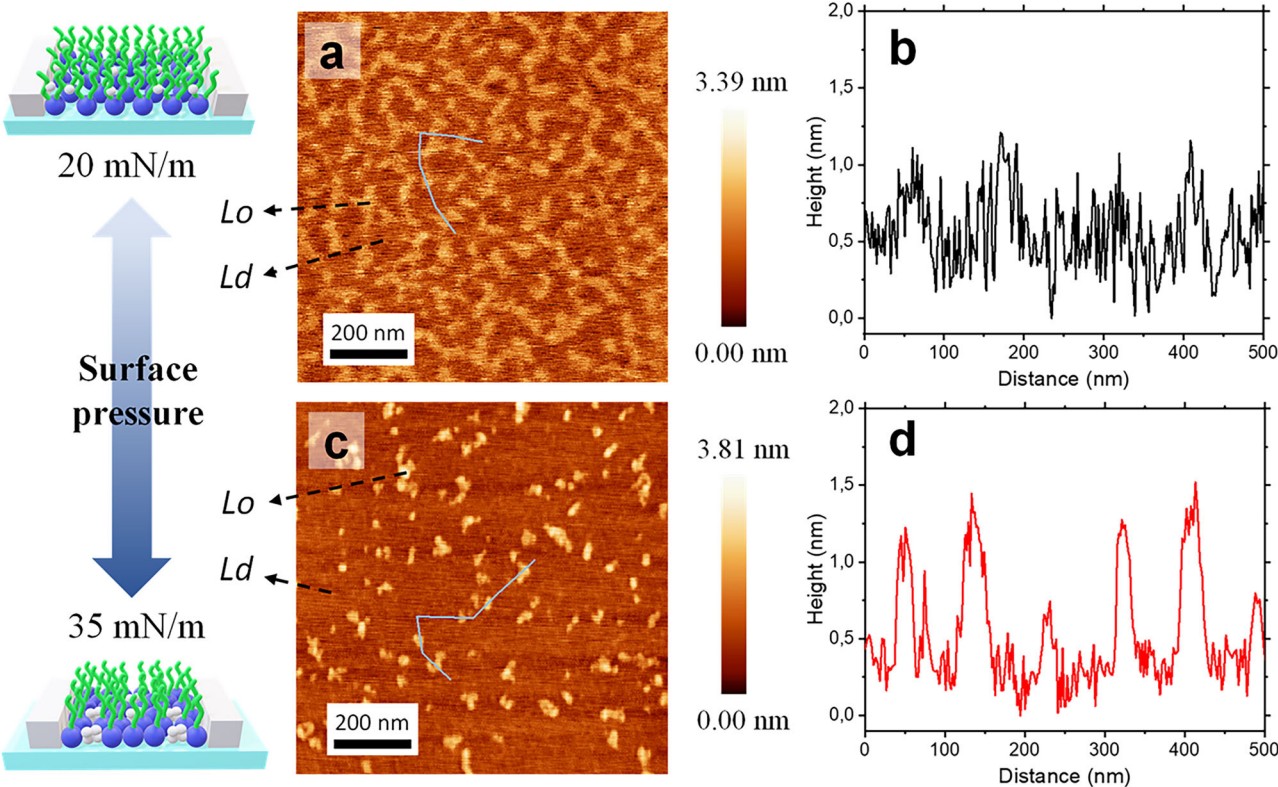

**Fig. 2 | Visualization of Lipid Organization Under Compression by AFM.** AFM topographical micrographs and corresponding topographical profiles of the pristine PM monolayer at $\Pi$ of 20 mN/m (**a**, **b**) and 35 mN/m (**c**, **d**). Schematic representations are also included for clarity. Source data for all graphs included in this figure is provided as a Source Data file.

compression indicates a disruption of the pre-existing spatial organization. This suggests that smaller, fragmented domains are energetically more favorable in this regime. The interplay between reducing free energy through tighter molecular packing and minimizing line tension, associated with the boundaries between the $Lo$ and $Ld$ phases, drives this organization. Smaller, randomly distributed $Lo$ domains reduce the cumulative line tension while maintaining the phase separation characteristic of the PM monolayer, achieving an optimal balance at higher surface pressures.

### Self-assembly of fusion peptides on PM monolayers revealed by AFM

To investigate the in-plane self-assembly of FPs and their impact on lipid organization, AFM was performed on LB monolayers of PM incorporating FPs. These FPs-PM monolayers, spread at the air/water interface, were transferred onto freshly cleaved mica at surface pressures of $\Pi = 20$ and 35 mN/m. Topographical AFM micrographs showing the different self-assembled FP structures at the PM LB interface are presented in Figs. 3 and 4.

### Straight and Stiff: Formation of Linear Peptide Fibers by FP1 on PM Monolayers.

At $\Pi = 20$ mN/m, FP1 self-assembles on the PM monolayer into long nanofibres, ranging from 100 nm to 700 nm in length and distributed across the monolayer (Fig. 3a, Supplementary Fig. 5A). The fibril heights ($H$), averaging $H = 1.0 \pm 0.2$ nm, are consistent with a peptide fibrillar arrangement in which the backbones align parallel to the surface plane and are in contact with the lipid acyl chain region of the PM monolayer. This conformation is further supported by molecular size calculations for FP1 in a β-strand structure, which predict dimensions of approximately 7.4 nm in length and 1.3 nm in height. Previous studies have shown that short β-sheet-rich peptides and

similar systems, such as HIV-1 FPs, form long, uniform fibers under uniaxial compression at the air-water interface[6,28].

Due to FP1's high hydrophobicity[11], the fibers mainly interact with the lipidic acyl chains of the PM. This is further supported by the topographical profiles in Fig. 3b, which show that the FP1 fibers are partially located in the acyl chains region of the PM rather than integrated within the lipid headgroups region. The average fibril height observed here ($H = 1 \pm 0.2$ nm) is notably smaller than the height of 2.0 nm reported for a pure FP1 monolayer[6]. This discrepancy confirms that FP1 is partially inserted into the PM. This likely reflects a balance of steric and noncovalent interactions dictated by FP1's amino acid sequence: hydrophobic residues intercalate into acyl chains, while more polar residues remain exposed to the aqueous phase. These findings indicate that, under LE-phase conditions, FP1 self-assembles into fibrillar structures at the PM interface. This behavior is governed by FP1's β-strand secondary structure and by the lipid packing state of the monolayer.

In contrast, at $\Pi = 35$ mN/m, where the PM adopts a C phase, the fibrillar structures observed at lower $\Pi$ disappear, resulting in a more homogeneous topography (Fig. 4a, b). AFM topographical images reveal only a minimal presence of peptide fibers compared with the pronounced fibrillar network at $\Pi = 20$ mN/m. This transition suggests that the increased uniaxial compression forces disrupt the ability of FP1 to maintain fibrillar assemblies within the lipid matrix, supporting the view that compression drives a shift from surface-associated fibers to peptides embedded within the monolayer. This integration results from steric hindrance and tighter lipid packing in the C phase, which restricts the space available for aggregation. Moreover, the disappearance of fibrillar structures may reflect changes in the balance of stabilizing forces: van der Waals and electrostatic interactions that dominate in the LE phase become less favorable under compression.

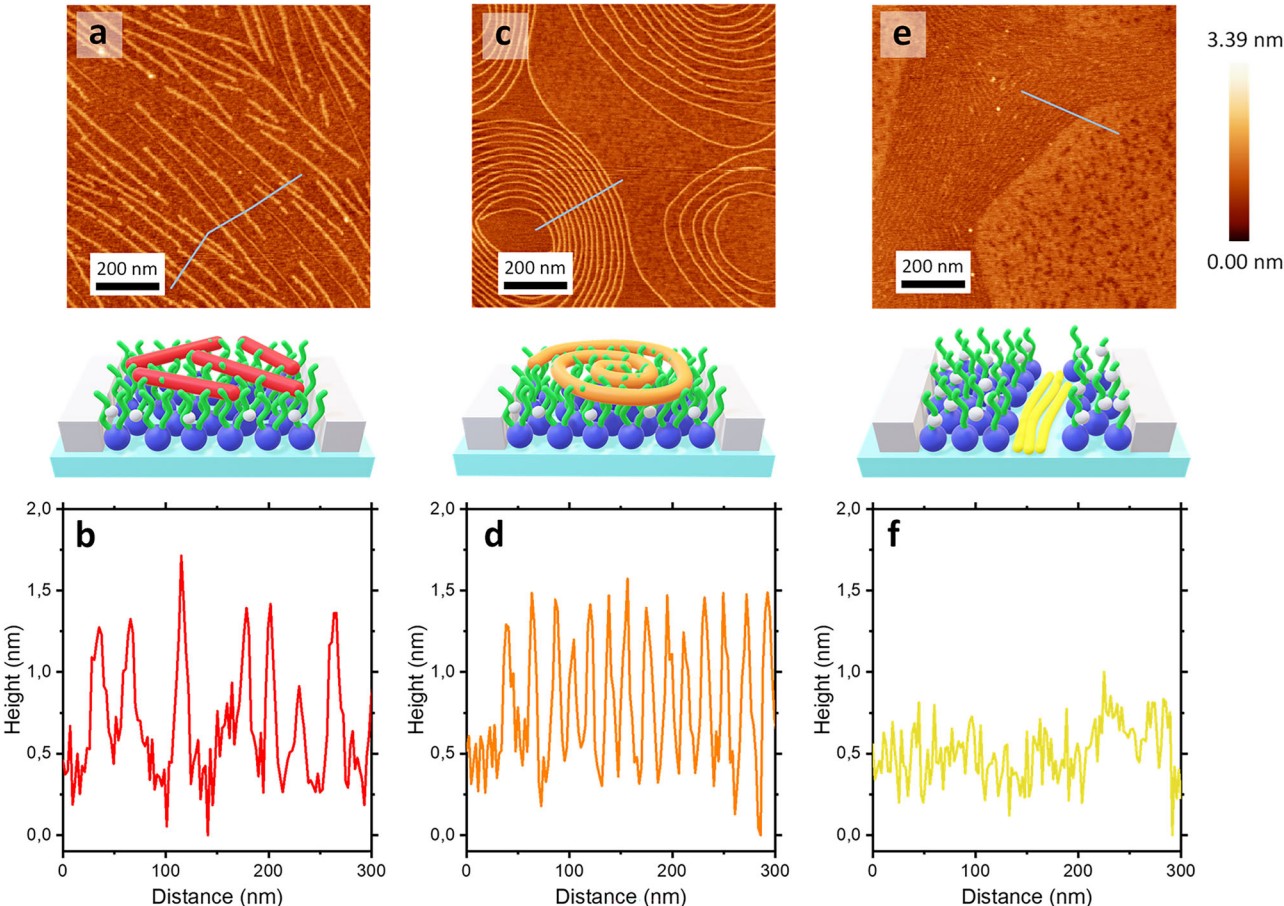

**Fig. 3 | AFM characterization of FP-PM Monolayers in a fluid phase.** AFM topographical micrographs and corresponding topographical profiles of PM-FP1 (**a**, **b**), PM-FP2 (**c**, **d**), and PM-FP4 (**e**, **f**) at 20 mN/m surface pressure onto mica substrate. Schematic representations of the corresponding AFM images are also presented for clarity. Source data for all graphs included in this figure is provided as a Source Data file.

These findings indicate that lipid packing not only dictates FP1's supramolecular organization but also modulates its functional role at the PM. By altering peptide–lipid interaction dynamics, the phase state of the PM directly impacts FP1 stability and assembly, with potential consequences for its biological activity.

**Twisting into spirals: FP2 forms into flexible nanofibers on PM monolayers.** AFM topographical micrographs of PM in the presence of FP2 at $\Pi = 20$ mN/m show elongated, flexible nanofibrils arranged into distinctive spiral assemblies (Fig. 2C and Supplementary Fig. 5b). This indicates cooperative interactions between fibers during FP2 self-assembly, potentially driven by specific intermolecular forces and/or steric compatibility, as also observed for FP1.

The spiral fibril heights for FP2 ($H = 1.2 \pm 0.2$ nm) indicate a fibrillar structure with backbones aligned parallel to the surface (Fig. 3d), interacting partially with the lipid acyl chain region of the PM monolayer, which is similar to FP1. This is consistent with molecular size calculations for FP2 in a β-strand conformation, which predict a length of ~8.1 nm and a height of ~1.5 nm. Previous studies have shown that β-sheet structures enhance FP2's ability to form spiral fibers at fluid interfaces[6]. Like FP1, FP2 predominantly interacts with the hydrophobic tail regions of the lipids in the PM, driven by hydrophobic interactions. However, unlike FP1, which is rich in hydrophobic residues (Leu, Ile, Val) and thus forms straight, rigid fibrils, FP2 contains a mixture of charged and polar residues (Lys, Asp, Gln). These can destabilize β-sheets through electrostatic repulsion and solvation effects, resulting in flexible fibrils that curl into spirals. The cysteine

residues in FP2 may also form disulfide bonds, further stabilizing the spiral morphologies observed at the PM interface.

Differences in fibril rigidity between FP1 and FP2 are evident in Fig. 3a, b. FP1 fibrils are long and straight, whereas FP2 fibrils show pronounced curvature, folding into spirals. Quantitative analysis of curvature (Supplementary Fig. 6) reveals a mean curvature angle ($\theta$) of $175 \pm 6°$ for FP1 fibrils, compared with $\theta = 164 \pm 7°$ for FP2 at $\Pi = 20$ mN/m. Persistence lengths ($L_p$) calculations, based on worm-like chain modeling (Supplementary Methods, Supplementary Fig. 7)[6], further highlight these differences: FP1 fibrils showed $L_p = 2.32 \pm 0.30$ μm, whereas FP2 exhibited a lower $L_p$ of $0.14 \pm 0.01$ μm, indicating greater flexibility. Remarkably, these values closely match those measured in the absence of lipids (FP1: $3.2 \pm 0.2$ μm and FP2: $0.12 \pm 0.03$ μm)[6], indicating that fibril stiffness and flexibility are largely intrinsic to peptide sequence rather than strongly influenced by the lipid environment at this surface pressure. Spiral assemblies of FP2 also exhibit a low bending rigidity ($\kappa \approx 8.3 \times 10^{-27}$ Nm²), calculated from elastic beam theory (see Supplementary Methods for details), compared with FP1 fibrils ($\kappa \approx 13 \times 10^{-27}$ Nm²), consistent with their flexible versus rigid morphologies.

At $\Pi = 35$ mN/m FP2 fibrils partially disappear from the PM surface (Fig. 4c, d). Instead, short fibres arranged in circular patterns remain on the PM, acting as remnants or "patch marks" of the previously observed spiral assemblies. Disassembly is probably driven by steric constraints and lipid compression, which destabilize the spirals and promote reorganization. This interpretation is supported by the reduction in molecular area per FP2, as shown by the Π-A isotherm

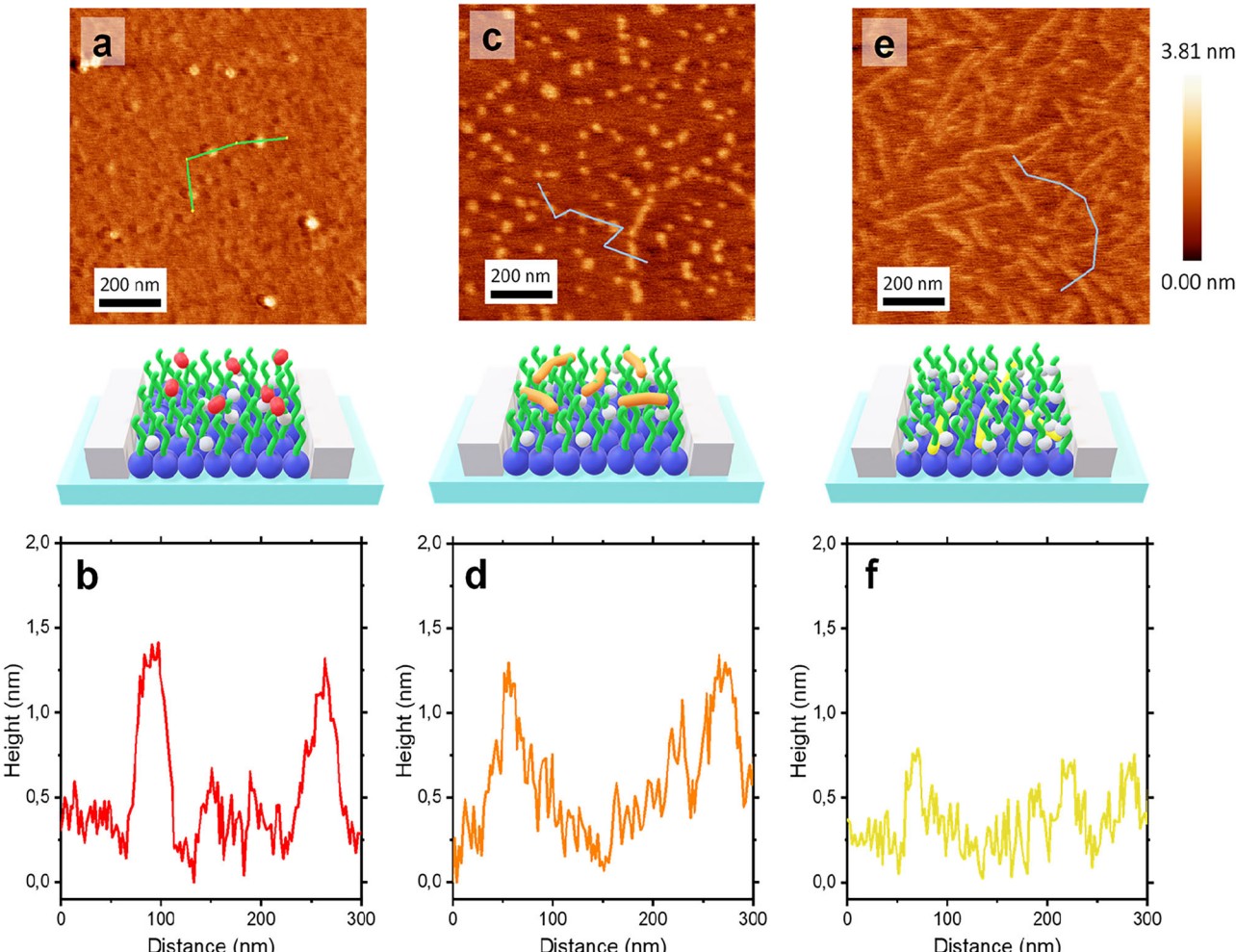

**Fig. 4 | AFM characterization of FP-PM Monolayers in a condensed phase.** AFM topographical micrographs and corresponding topographical profiles of PM-FP1 (**a**, **b**), PM-FP2 (**c**, **d**), and PM-FP4 (**e**, **f**) at 35 mN/m surface pressure onto mica substrate. Schematic representations of the corresponding AFM images are also presented for clarity. Source data for all graphs included in this figure is provided as a Source Data file.

(Fig. 1d). Notably, in the absence of lipids, the spirals remained intact, suggesting that lipid interactions and the associated steric hindrance are key factors driving spiral disassembly under compression[6].

**Electrostatic stabilization and structural resilience of FP4 fibrils on PM monolayers.** The strong affinity of FP4 for the PM, characterized by a binding free energy of 9.7 kcal/mol, is primarily driven by electrostatic interactions between its overall positive charge and the negatively charged phospholipid headgroups[11]. This affinity is confirmed by the Π-A isotherms in Fig. 1d, which show significant perturbation of the lipid monolayer upon FP4 incorporation.

AFM micrographs (Fig. 3e and Supplementary Fig. 5C) reveal that FP4 induces phase separation in the PM, resulting in two different regions: one enriched in parallel-aligned peptide fibers and the other composed of phospholipids and cholesterol, with visible *Lo* and *Ld* domains. The negligible difference in height between the lipidic domains and the FP4 fibers (Fig. 3f) suggests complete segregation, with the FP4 fibers positioned between lipid domains rather than integrated within them, unlike FP1 and FP2.

The assembly of FP4 into short, linear fibrils arises from its β-sheet secondary structure in the presence of membranes, as confirmed by circular dichroism[11]. The ordered fibrillar organization locally disrupts lipid packing, producing peptide-rich domains. Glycine-rich segments in FP4 enhance backbone flexibility, facilitating extended conformations and tight β-sheet packing into parallel fibrils. Aromatic residues further stabilize this alignment through π-π stacking interactions. Together, these features explain FP4's ability to form well-ordered parallel superstructures, in contrast to FP1 and FP2, which lack comparable glycine-rich sequences and aromatic stacking capacity.

Notably, FP4 fibrils persist within the PM monolayer at $\Pi$ = 35 mN/m, in contrast to FP1 and FP2, which disassemble under compression. AFM topographical images (Fig. 4e, f) reveal that FP4 fibrils remain integrated into the PM monolayer under these conditions, with uniaxial compression promoting the merging of previously segregated peptide-rich and lipid-rich domains.

**Study of the impact of FP on PM adhesive properties by AFM.** To further validate the structural observations, the adhesive forces of PM and FP monolayers were measured using AFM in QNM mode. Supplementary Fig. 8 presents adhesive force maps for the samples studied during this work at $\Pi$ of 20 and 35 mN/m.

At $\Pi$ = 20 mN/m, the pristine PM exhibited adhesive forces of ~1.6 nN, which remained largely unchanged upon addition of FPs, suggesting that peptides are more likely segregated from than integrated into the PM. For FP1 and FP2 (Figures S8B, C), fibers are clearly visible in the adhesive force maps, displaying minimal adhesion due to their hydrophobic character. This finding supports the AFM topography results, confirming that FP fibers localize primarily within the

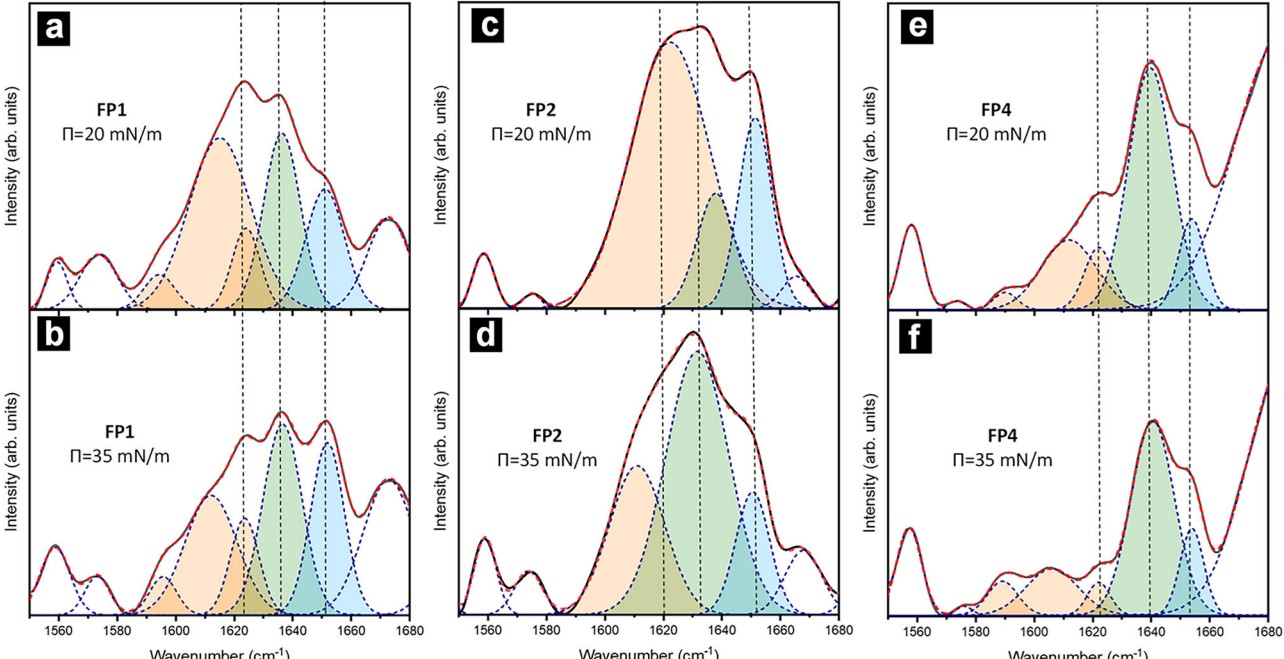

**Fig. 5 | Secondary structure decomposition from LDIR spectra. a–f** LDIR spectra of the samples studied during this work. In each spectrum, the deconvolution of the amide I band is shown into its main components: β-sheet (orange), random coil (green), and α-helix (blue). Source data for all graphs included in this figure is provided as a Source Data file.

aliphatic tail region of the PM monolayer. In contrast, FP4 (Supplementary Fig. 8D) displayed extended peptide-rich domains corresponding to the large fiber regions observed in Fig. 3e. Together with the negligible height differences in topography, this indicates that FP4 segregates into distinct domains rather than mixing with the lipid matrix.

At $\Pi$ = 35 mN/m, the aggregation of SM and cholesterol suggested by topography was confirmed by adhesive force imaging of the pristine PM (Supplementary Fig. 8E), with low-adhesion areas corresponding to hydrophobic sterol- and sphingolipid-rich domains. For FP1 (Supplementary Fig. 8F) and FP2 (Supplementary Fig. 8G), no fibers were detected, indicating peptide insertion into the membrane and disassembly of previously observed fibrils. Consistent with GIXD results, cholesterol and SM aggregation were also absent, showing that FPs prevent segregation of these components. In the case of FP4, increased packing under compression promoted the formation of extensive, homogeneous peptide domains integrated within the PM (Supplementary Fig. 8H).

**LDIR Analysis of Surface-Pressure Effects on FP Secondary Structure.** Previous studies have established that peptide fiber formation is closely linked to secondary-structure content[29]. We hypothesize that, in FPs, transitions in secondary structure act as molecular switches, converting mechanical stimuli into membrane-remodeling actions during viral entry. To test how $\Pi$ influences these transitions, and consequently fiber morphology in the PM environment, mixed PM-FP (FP1, FP2, and FP4) monolayers were analyzed by LDIR spectroscopy in reflection mode. Figure 5 shows the amide I spectra (1600-1700 cm$^{-1}$) recorded at two surface pressures ($\Pi$ = 20 and 35 mN/m).

At $\Pi$ = 20 mN/m, all PM-FP monolayers exhibited well-defined amide I bands. Spectral deconvolution revealed that FP2 contained the highest fraction of β-sheet structure, characterized by a dominant absorption at ≈1623 cm$^{-1}$ (ν⊥ mode) and a weaker band at ≈1680 cm$^{-1}$ (ν∥ mode), consistent with antiparallel β-sheet packing (Fig. 5a). Because β-sheet assemblies are relatively flexible, FP2 fibers could adopt curved morphologies, forming spiral structures as observed in

the AFM images (Fig. 3c). In contrast, FP1 exhibited a predominantly α-helical secondary structure, as indicated by a strong absorption near 1650 cm$^{-1}$, with additional random coil and β-sheet contributions (Fig. 5c). This structural profile is consistent with the formation of straight, rigid fibrils (Fig. 3a), reflecting the greater stiffness typically associated with α-helical backbones. FP4 exhibited a predominantly disordered secondary structure, with amide I spectra dominated by random coil features (Fig. 5e). Nonetheless, the aligned fibrillar domains observed by AFM (Fig. 3e) suggest that partial ordering, possibly involving short β-sheet segments stabilized by its glycine-rich motifs, may be sufficient to nucleate and stabilize lateral fiber growth at the membrane interface.

Upon increasing $\Pi$ to 35 mN/m, the β-sheet component of the amide I band diminished markedly for FP1 and FP2, whereas the α-helical signal remained comparatively stable. This selective loss of β-sheet absorbance correlates with the near disappearance of fibrillar features in AFM topographical micrographs (Fig. 4a, c), indicating that compression disrupts the hydrogen-bonding network required for extended β-sheet assembly. In contrast, FP4 exhibited minimal change in its amide I profile (Fig. 5f), consistent with AFM observations that FP4 fibrils remain integrated within the membrane at higher pressures (Fig. 4e).

## Discussion
### Biological impact and proposed mechanism of fusion
The membrane fusion process of SARS-CoV-2 (Fig. 6a) emerges from the coordinated activity of its fusion peptides, which engage the plasma membrane with complementary functions[14]. In contrast with the conventional view of monomeric or loosely associated FPs, our data show that these short motifs do not act independently but form an integrated system in which anchoring, energy storage, lipid remodeling, and pore destabilization are sequentially combined to drive viral entry.

A long-standing question in FP biology is whether the oligomeric forms observed in vitro represent physiologically relevant intermediates. In the viral context, only a subset of Spikes is likely to engage

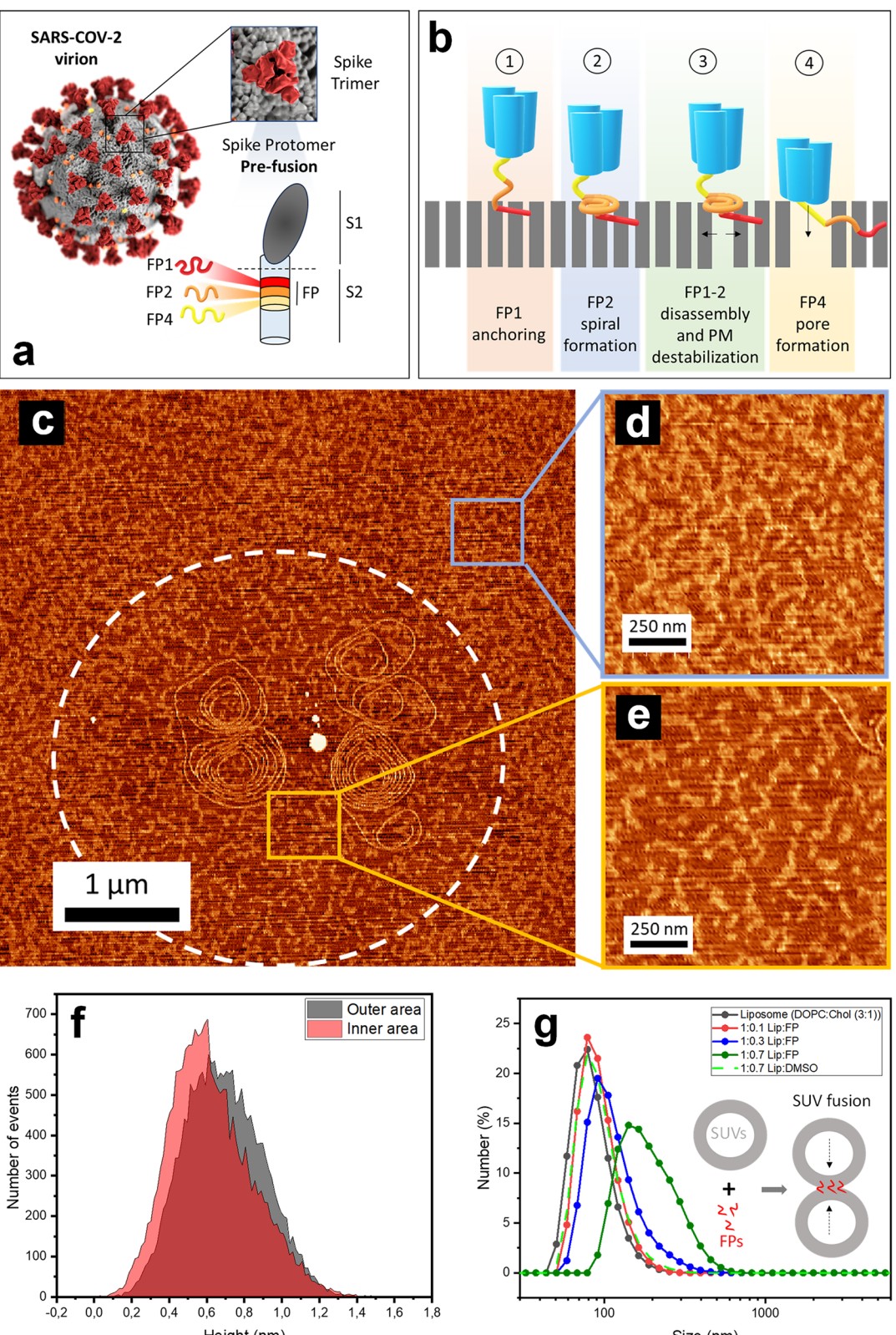

a given membrane-membrane interface, meaning that the peptide densities achieved here represent mechanistic upper limits rather than exact physiological stoichiometry. The cooperative assemblies we describe should therefore be interpreted as illustrative models of how fusion motifs can remodel membranes when locally enriched, rather than literal reconstructions of the viral fusion interface.

In this study, SARS-CoV-2 FPs displayed markedly different self-assemblies upon contact with lipid membranes, shaped by their charge, hydrophobicity, and secondary structure. FP binding altered monolayer area per molecule (Fig. 1d), surface topographies and mechanics (Figs. 3 and 4), and excluded cholesterol-rich domains (Fig. 1g, h). These effects suggest that, although the intact Spike

**Fig. 6 | Cooperative mechanism of SARS-CoV-2 fusion peptides. a** Schematic of the SARS-CoV-2 virion and the Spike protein in its prefusion conformation, highlighting the location of fusion peptides FP1 (red), FP2 (orange), and FP4 (yellow) within the S2 subunit. Illustration created by the U.S. Centers for Disease Control and Prevention (CDC) and obtained from the CDC's free public media repository. This graphic is provided solely for visualization purposes and is not derived from experimental data. **b** Stepwise representation of their cooperative functions during membrane fusion. (1) FP1 inserts into lipid tails, anchoring the viral complex to the host membrane. (2) FP2 forms spiral fibrils that act as elastic reservoirs to accumulate mechanical stress. (3) The contiguous FP1-2 tandem couples anchoring with curvature generation; spiral disassembly releases stored energy and destabilizes the plasma membrane. (4) FP4 penetrates into the bilayer under compression, disrupting lipid packing and promoting pore expansion. **c** AFM micrograph of the FP1-FP2 hybrid interacting with PM. Higher magnification AFM micrographs of the PM in areas far away (**d**) and in close contact (**e**) to the FP interaction point and corresponding height histograms (**f**). DLS of the interaction of DOPC: cholesterol liposomes and FP1-FP2 hybrids (**g**). Source data for all graphs included in this figure is provided as a Source Data file.

engages ACE2 within cholesterol-rich lipid rafts[30,31], the cleaved FPs disrupt these domains, increasing local fluidity to facilitate fusion-pore formation. It should also be noted that in the native Spike, FP1, FP2, and FP4 are contiguous motifs within the S2 subunit, whereas here they were studied individually and at relatively high concentrations. These conditions likely enhance supramolecular assembly, but they also allow a reductionist dissection of the distinct functions contributed by each motif.

FP1 establishes the first stable contact with the host membrane (Step 1 at Fig. 6b). At moderate packing ($\Pi = 20$ mN/m), it organizes into long, rigid fibrils consistent with β-sheet/α-helical structures observed by LDIR spectroscopy, and AFM profiles reveal insertion into the acyl chain region. Compression to $\Pi = 35$ mN/m promotes even deeper insertion, turning FP1 into a molecular anchor that secures the viral complex without markedly perturbing the lateral organization of lipid phases. This anchoring role provides a fixed scaffold at the membrane interface from which the other peptides act.

FP2 exhibits a markedly different behavior under identical conditions. Its sequence composition and β-sheet propensity drive the formation of elongated, spiral fibrils that remain partially associated with lipid acyl chains (Step 2 at Fig. 6b). These spirals exhibit lower persistence length ($\approx 0.14$ μm) compared with the rigid FP1 fibrils ($\approx 2.3$ μm), indicating a high degree of flexibility. The resulting curvature stores elastic energy, estimated at ~17 kcal/mol ($\approx 25$ kBT) per spiral (see Supplementary Methods), values approaching the ~50 kcal/mol required to overcome the hemifusion barrier. Under compression ($\Pi = 35$ mN/m), these assemblies disassemble, releasing their stored stress precisely when membranes are forced into close apposition (Step 3 at Fig. 6b). Such loaded-spring behavior strongly parallels ESCRT-III Snf7 filaments, which generate curvature and buckling during cellular membrane remodeling[32]. In this way, FP2 acts as an elastic reservoir that supplies the mechanical work necessary to destabilize the bilayer and promote stalk formation.

FP4 completes the fusion machinery by destabilizing the membrane at later stages (Step 4 at Fig. 6b). With a binding free energy of ~9.7 kcal/mol toward negatively charged headgroups, it shows strong affinity for phospholipids and responds to membrane packing. At low surface pressures, AFM and adhesive force mapping reveal fibrils segregated from lipid domains, while at $\Pi = 35$ mN/m, a surface pressure that reflects the rise in lateral lipid packing expected during the early stalk-formation stage of membrane fusion, FP4 transitions into β-sheet-rich parallel fibrils integrated into the bilayer. This insertion perturbs bilayer continuity, providing a mechanism for pore expansion, something already probed with cryo-EM[2]. The shift from surface binding to insertion mirrors the sequence of viral fusion itself, in which initial headgroup association precedes penetration into the hydrophobic core.

Together, these findings support a model in which FP1 secures the viral complex, FP2 assembles into spirals that accumulate and release elastic stress to deform the bilayer, and FP4 penetrates into the membrane under compression to destabilize and expand pores. We emphasize that this framework should be considered a mechanistic model derived from reductionist peptide-membrane systems, not a literal reconstruction of Spike activity in vivo. Accordingly, the extrapolation to the physiological process of SARS-CoV-2 entry must be made with caution. This minimal but efficient system concentrates mechanical forces at the fusion site, converting peptide binding energy into targeted bilayer remodeling. Beyond providing a molecular explanation for Spike-mediated fusion, these results demonstrate how simple peptide motifs can be organized into cooperative assemblies that capture, store, and release mechanical energy with high spatial and temporal precision, suggesting new strategies for the design of responsive peptide-based nanomaterials.

## Tandem FP1-2: coupling anchoring and curvature to validate the mechanism

The contiguous arrangement of FP1 and FP2 in the S2 subunit (residues 816-856) suggests that their functions are not independent but are designed to act in concert during fusion. To test this hypothesis, we engineered a tandem FP1-2 construct, reasoning that the combination of FP1's anchoring ability with FP2's curvature-inducing spirals would recapitulate the cooperative mechanism proposed above.

AFM imaging (Fig. 6c) confirmed the formation of extended supramolecular spiral assemblies, reminiscent of those formed by FP2 alone (Fig. 3c). These spirals exhibited greater persistence length and bending stiffness than FP2-only fibrils (Supplementary Table 1), likely due to FP1-derived rigidity. Importantly, FP1-2 spirals drove localized lipid remixing: AFM images revealed a reduction in $Lo/Ld$ phase contrast near peptide assemblies. High-resolution scans (Fig. 6d, e) showed that regions adjacent to FP1-2 structures contained fewer $Lo$ domains, while more distant areas retained higher $Lo$ content. Cholesterol-enriched $Lo$ regions thus appeared at lower density relative to the surrounding $Ld$ matrix, indicating peptide-induced lateral lipid redistribution. This local disruption of ordered domains likely softens otherwise rigid regions and promotes interfacial heterogeneity favorable for membrane deformation and fusion. AFM height histograms further supported this effect: membrane thickness variance decreased by ~25% in FP1-2 contact zones (Fig. 6f), consistent with lipid-phase homogenization.

To validate monolayer findings in more physiological architectures, FP1-2 was incubated with DOPC:cholesterol small unilamellar vesicles (SUVs). Dynamic light scattering confirmed homogeneous vesicles (diameter = $80 \pm 3$ nm, PDI < 0.1; Fig. 6G). Even at sub-stoichiometric ratios (liposome:peptide = 1:0.7), FP1-2 triggered rapid vesicle expansion to 200 nm (150% increase; Fig. 6g). This >2.5-fold increase demonstrates that the tandem construct promotes vesicle aggregation and fusion, reflecting strong synergistic remodeling activity. Importantly, such peptide-driven remodeling mirrors cellular observations where even minimal levels of SARS-CoV-2 Spike protein on viral particles are sufficient to induce syncytia formation[33]. Together, these results reinforce the view that FPs act synergistically as a minimal fusogenic module capable of driving large-scale membrane reorganization.

In summary, this study shows how SARS-CoV-2 FPs interact with membrane models to drive early steps of viral entry. Each peptide displayed distinct behaviors: FP1 formed rigid fibrils that anchor into lipid tails, FP2 built flexible spirals that store and release mechanical energy, FP4 created stable parallel fibrils that persist under

compression, and the FP1-2 tandem combined anchoring with curvature induction. Together, they disrupted cholesterol-rich domains, thinned membranes, and increased fluidity, changes that lower the energy needed for fusion intermediates such as hemifusion stalks. These results support a model in which clustering of Spike trimers concentrates FPs, allowing cooperative action to bend and destabilize host membranes, while recognizing that the assemblies observed here arise from isolated peptide fragments studied in reductionist systems. The elastic spiral assemblies of FP2, similar to ESCRT-III protein spirals, highlight a shared strategy of storing and releasing mechanical stress to remodel membranes. Beyond explaining viral fusion, these findings provide design rules for bio-inspired nanomaterials that harness simple peptide assemblies to control membrane structure and mechanics.

## Methods

### Fusion peptides

The fusion peptides were synthesized and purified by GenScript (Amsterdam, The Netherlands). Stock solutions of each peptide in DMSO (dimethyl sulfoxide) were used for all the experiments reported here. The following peptides were investigated: FP1 (SARS-CoV-2 816-SFIEDLLFNKVTLADAGFIKQY- 837); FP2 (SARS-CoV-2 835-KQYGDCLG-DIAARDLICAQKFN- 856); and FP4 (SARS-CoV-2 885-GWTFGA-GAALQIPFAMQMAYRFNGI- 909). See Supplementary Table 2 for further details of the peptides.

### Hybrid FPs-lipid mixture solutions preparation and biomimetic PM formulation

Outer leaflet of eukaryotic plasma membranes was prepared in the form of Langmuir monolayers as detailed elsewhere[34,35]. Briefly, phosphatidylcholine (PC), phosphatidylethanolamine (PE) and phosphatidylserine (PS) extracted and purified from perdeuterated and hydrogenous *Pichia pastoris* cell cultures were mixed by cholesterol and egg yolk sphingomyelin (SM) purchased in powder form from Sigma-Aldrich. The composition in molar ratio was PC 0.2, PE 0.11, PS 0.06, cholesterol 0.5, SM 0.13 (Supplementary Table 3). Lipid stock solutions were prepared in chloroform stabilized with ethanol (purity 99.8%; Sigma Aldrich, St. Louis, MO, USA), and stored at -20 °C. Mixed solutions of lipids and peptides (PM-FP) were prepared by adding 5% moles (*i.e.*, 1:20 FP:lipid) of FP to the PM solutions. The final lipid concentration used in all solutions was maintained at 0.2 mg/mL.

### Langmuir trough experiments

Surface pressure ($\Pi$)-area ($A$) isotherms of hybrid PM-FPs monolayers were acquired using a LB trough (model G2, KIBRON, Helsinki, Finland). Surface pressure variation was recorded using a Wilhelmy plate made of filter paper. The temperature was maintained at $21.5 \pm 0.5$ °C. The trough was filled with de-ionized water (Milli-Q, Millipore; resistivity higher than 18 MΩ·cm), and freshly cleaved mica substrates were vertically immersed using a dipper. A 0.2 mg/mL lipid mixture (PM), in chloroform, was spread over the subphase using a Hamilton microsyringe, achieving an initial surface pressure of 2 mN/m. After chloroform evaporation (20 min) surface pressure variations during compression were recorded using the Wilhelmy plate. Once the target surface pressure (20 and 35 mN/m) was reached, the interfacial assemblies were transferred onto mica substrate (discs with of 12 mm diameter). Similar protocol was followed for 0.2 mg/ml PM + 5% FP1, 0.2 mg/ml PM + 5% FP2, 0.2 mg/ml PM + 5% FP4, 0.2 mg/ml PM + 5% FP1-2 monolayers.

### Area per molecule calculation by neutron reflectometry

Quantification of the membrane composition was done by neutron reflectometry (NR) exploiting the low-$q_z$-range approach, *i.e.*, by collecting data at a limited $q_z$-range, $0.01 \text{ Å}^{-1} < q_z < 0.03 \text{ Å}^{-1}$ ($\lambda$ from 4.5 to 13 Å), which allows a relatively fast acquisition time. NR experiments were performed on the time-of-flight reflectometer FIGARO at the Institut Laue-Langevin, Grenoble (France)[36]. Only the lowest angle of incidence ($\theta = 0.6°$) was used for the experiments, and a wavelength resolution of 7% d$\lambda$/$\lambda$. The raw time-of-flight experimental data were normalized with respect to the incident wavelength distribution and the efficiency of the detector yielding the resulting R($Q_z$) profile using COSMOS[37]. A one-layer model of the PM and PM-FP monolayers, respectively (Supplementary Table 4) was used to determine the area per molecule ($A$) of phospholipids and fusion peptides at different values of $\Pi$ following previous studies (Supplementary Fig. 1)[38]. The are per molecule is given by $A = \sum b_i / \rho_i \cdot t$, where $\sum b_i$ and $\rho_i$ are the scattering length and the scattering length density of each component, respectively, and t is the film thickness (Supplementary Table 5). The data analysis was performed by fitting the product of $\sum \rho_i$ and t. This was done for deuterated PM monolayers in presence/absence of FPs in $D_2O$:$H_2O$ mixture (8:92% v/v) subphase. In the fitting process performed using REFNX[39], the interfacial roughness was fixed to 3 Å, which is consistent with the capillary waves of the subphase[40,41].

### Atomic force microscopy

The monolayer formed by PM in the absence and presence of FP were imaged at the air-solid interface by AFM. All images were obtained with a multimode AFM equipped with a Nanoscope V controller (Bruker). The AFM was operated in Peak Force mode, at room temperature[42]. Silicon nitride cantilevers (model: PNP-DP, NanoAndMore GmbH, Germany) with force constant 0.48 N/m, length 100 µm and a resonant frequency of 67 kHz, were used for scanning. The images were taken at a scan rate of 1 Hz and 512 × 512 pixels. The images were acquired with the Nanoscope Software. They were topologically flattened and analysed by using WSxM software[43]. Nanomechanical properties were determined in the peak force QNM mode of AFM. The advantage of using peak force QNM is that the adhesive force between the sample and tip and the magnitude of sample deformation caused by the tip are both directly measured in real-time. The cantilever was calibrated using the standard hard surface (sapphire) and images were analysed using NanoScope Analysis software (V1.9, Bruker).

### Grazing-Incidence X-ray diffraction (GIXD)

Experiments were performed at ID 10 beamline at the European Synchrotron Radiation Facility (ESRF), Grenoble, France, with X-ray energy 22 keV and a beam size of $25 \times 13$ µm$^2$. This technique has been extensively described elsewhere[44,45]. Here, an in-house, setup consisting of a PTFE Langmuir trough equipped with a single moveable barrier, was used for GIXD experiments. The experiment was performed at two selected pressures (20 and 35 mN/m) and at a constant subphase temperature of $21 \pm 0.5$ °C. To minimize the background scattering, the trough was isolated in a Kapton box, and the inside atmosphere was saturated in He (oxygen level < 0.2%). Different areas of the trough were exposed to the X-ray beam in each particular experiment to avoid damage to the sample[44].

For GIXD experiments, Langmuir monolayers were irradiated at an incidence angle of $\theta = 0.1233°$, 80% below the critical angle of pure water. GIXD 2D contour profiles of the scattered intensity were acquired using a double linear detector (Mythene 2 K) mounted behind a vertically oriented Sollers collimator with an in-plane angular resolution of 1.4 mrad. Diffracted intensities were detected as a function of the X-ray momentum transfer component perpendicular, $q_z$, and parallel to the air/water interface, $q_{xy} = (4\pi/\lambda)\sin 2\theta_{xy}/2$. GIXD peaks were obtained by the integration of the 2D profiles along $q_z$ to obtain the Bragg peaks using in-house scripts developed at ESRF ID10. These data were fitted using the Lorentzian function in order to obtain the peak position and its full-width half maxima (FWHM). In-plane coherence length ($L_{xy}$) along the crystallographic direction was determined using the Scherrer formula: $L_{xy} = (0.9 \times 2\pi)/\text{FWHM}$.

### Laser direct infrared (LDIR)

Spectra were acquired using an IR Imaging systems IR Agilent LDIR 8700 on Au-coated silicon substrates, fabricated following the same methodology as for the mica surfaces. To minimize potential interference from water absorption bands at 1630-1650 cm$^{-1}$, samples were stored under vacuum for 48 h prior to measurements.

### Preparation of DOPC: cholesterol liposomes and characterization by dynamic light scattering

1,2-Dioleyl-sn-glycero-3-phosphocholine (DOPC; Avanti Polar Lipids, Alabaster, USA): Cholesterol (Merck) liposomes were prepared by dissolving the lipids in chloroform, followed by solvent evaporation to form dry lipid films and overnight vacuum desiccation. The films were rehydrated in buffer to yield multilamellar vesicles, which were subsequently processed by repeated freeze–thaw cycles, tip sonication, and extrusion through 100 nm polycarbonate membranes to obtain unilamellar vesicles. Vesicle size and homogeneity, in the absence and presence of FP1-2 hybrid, were assessed by dynamic light scattering (Zetasizer Nano ZS90, Malvern Instruments, UK) using a low-volume disposable cuvette. DLS measurements were performed at $21.0 \pm 0.5\,°C$ in quasi-backscattering configuration (scattering angle, $\theta = 173°$) using the red line of a He-Ne laser working at a wavelength $\lambda = 632.8$ nm. Each measurement consisted of 12 scans and was repeated three times to ensure reproducibility. The presented data represent the average values (± standard deviation) obtained from these measurements.

### Statistics and reproducibility

Surface pressure and molecular area values (Fig. 1d, e) were obtained from neutron reflectivity fits, with uncertainties corresponding to the standard error of the fitted area parameter, while shaded regions represent the instrumental systematic error of $\pm 1\,mN\,m^{-1}$, which exceeded the variability across three independent replicates ($n = 3$). Compressional elasticity values (Fig. 1f) were determined from the slope of the surface pressure-area isotherms, and uncertainties reflect the standard error from linear regression, including propagated instrumental error in surface pressure ($\pm 1\,mN\,m^{-1}$). Coherence lengths were calculated from grazing-incidence X-ray diffraction (GIXD) patterns by fitting the Full Width at Half Maximum (FWHM) of each Bragg peak using a Gaussian function; the non-linear least squares fitting routine provided standard errors for the FWHM, which were propagated to each coherence length ($L_{xy}$). Due to the high signal-to-noise ratio of the GIXD data, the resulting uncertainties were smaller than the experimental symbols shown in Fig. 1i, j. For the atomic force microscopy (AFM) analysis, a total of fifteen large-area ($10 \times 10\,\mu m^2$) images were acquired per condition from three independent samples ($n = 3$; five images per sample) to ensure statistical robustness and assess morphological homogeneity. The AFM images presented in Figs. 2–4, and 6 are representative of the predominant features consistently observed across all replicates and regions, and no anomalous or non-characteristic features were included in the analysis.

### Reporting summary

Further information on research design is available in the Nature Portfolio Reporting Summary linked to this article.

## Data availability

Source data are provided with this paper. In addition, the data that support the findings of this study is available online through Zenodo at 10.5281/zenodo.17408449. Source data are provided with this paper.

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

## Acknowledgements

The authors thank the Institute Laue-Langevin (ILL) for allocation of neutron beam time on FIGARO (https://doi.org/10.5291/ILL-DATA.DIR-215) and the European Synchrotron Radiation Facility (ESRF) for provision of synchrotron radiation facilities in beamline ID10 under proposal number LS3159 (https://doi.org/10.15151/ESRF-ES-883953841). We acknowledge O. Konovalov for his valuable assistance during the GIXD experiments. A.M. acknowledge the financial support from MCIN/AEI/10.13039/501100011033 under grants PID2021-129054NA-I00 and PID2024-157988NB-I00, as well as the financial support from the Department of Education of the Basque Government under grant PIBA_2023_1_0054 and from the IKUR Strategy under the collaboration agreement between Ikerbasque Foundation and Materials Physics Center. E.G. acknowledge the financial support from MCIN/AEI/10.13039/501100011033 under grant PID2023-147156NB-I00 and UCM under grant PR12/24-31566 (Ayudas para la financiación de proyectos de investigación UCM 2023). AAF is grateful for support from the Provincial Council of Gipuzkoa under the program Fellow Gipuzkoa, the "Ramón y Cajal" Program, Grant No. RYC2024-050339-I funded by MICIU/AEI/10.13039/501100011033 and ESF +, as well as from the project PID2024-157277NA-I00 funded by MICIU/AEI /10.13039/501100011033 and FEDER, UE. The authors thank the technical and human support provided by SGIker (EHU/ERDF, EU) in conducting the LDIR measurements.

## Author contributions

A.M. and A.A.F. contributed to conceptualization, supervision, methodology, investigation, formal analysis, resources, data curation, writing - original draft, project administration, and funding acquisition. N.R.Z. contributed to conceptualization, writing - review & editing. N.P. was involved in investigation, formal analysis, data curation, and writing - review & editing. A.S, B.R., K.B., V.L., and E.G. all contributed to investigation and writing - review & editing.

## Competing interests

The authors declare no competing interests.
