## [Transparent Peer Review file · Nature Communications]

Insights into the Self-Assembly and Interaction of SARS-CoV-2 Fusion Peptides with Biomimetic Plasma Membranes

Corresponding Author: Dr Alberto Alvarez Fernandez

Version 0:

Reviewer comments:

Reviewer #1

(Remarks to the Author)

This article examines the self-assembly of fusion peptides (FPs) derived from the S2 subunit of the SARS-CoV-2 Spike (S) protein and their interactions with biomimetic plasma membrane (PM) models. While the study provides valuable insights into the self-assembly behavior of these peptides in controlled experimental conditions, there appears to be a disconnect between the abstract and the actual content of the article.

The abstract presents the study as one focused on viral entry mechanisms, which would be directly relevant to the biological context of SARS-CoV-2 infection. However, upon reading the article, it becomes clear that the study is more centered on the self-assembly of materials, rather than exploring biological mechanisms in a way that mimics the conditions encountered during viral infection. The focus is more on how these fusion peptides behave in isolation and under specific experimental setups (such as biomimetic membrane models), rather than their role in the actual context of viral infection, where viral S protein density on the viral surface would play a significant role in the interactions with host membranes.

Thus, I believe the manuscript would benefit from a clearer connection to viral infection biology and how the self-assembly of these peptides could inform our understanding of viral entry. While the technical aspects of peptide self-assembly and their interactions with lipid monolayers are interesting and well-executed, the article does not sufficiently address how these findings relate to the broader biological context of SARS-CoV-2 infection. The article would be more aligned with biological research if it more directly connected the observed phenomena to infection-related processes, such as how the density of S protein on the viral surface could influence the formation of these peptide assemblies and, in turn, affect viral fusion mechanisms.

Reviewer #2

(Remarks to the Author)

The paper "Insights into the Self-Assembly and Interaction of SARS-CoV-2 Fusion Peptides with Biomimetic Plasma Membrane Models" by Pavar et al. addresses questions regarding fusion peptides of SARS-CoV-2 with phospholipid membranes applying AFM and scattering techniques (x-ray and neutrons). A subdomain in the spike protein is crucial for the very successful attack of cell membranes.

The detailed structure of the fusion peptide (FP) with the biological plasma membrane model (PM) is investigated and different morphologies and structural assemblies are identified in the membrane.

The results are claimed to be relevant for drug delivery, bio sensing and membrane engineering.

The fibre formation of the FPs depending on the type of FP (1,2,4) and on the membrane surface pressure and is an interesting investigation.

Nonetheless I could not find a satisfactory answer how the described and shown FP effects are related to a SARS CoV-2 membrane fusion, since a spike protein in my understanding has only one of the FPs and not the whole fibre of some 100s of nm length as in Fig 3. Where is the connection? On the other hand, it is suggested that the fibres have some drug delivery relevance, I did not really get how this could be the case.

All in all the study is sound and interesting, the claimed relevance for membrane fusion for the virus entering the cell as well as for other biomedical applications needs to be explained much better for this journal.

Some smaller remarks:

Fig 1: G,H: could be mentioned that these are GIXD data (not neutrons).

Fig 1 I,J: L_{xy} in which units? Nm? Angstrom? The red dots vary strongly from $L_{xy}=40$ (PM, +FP4) to ~ 1 (+FP1,2). This comes from cuts from the Figures S3, right? Especially A,C,D seem to look quite similar, it is astonishing to me that the FWHM of the second peak should be so different (for B the noise seems to be larger, that it is rather difficult to assess a FWHM at all?). Also Fig.1H, FP2 has a peak which seems to be missed by the fit.

Are the GIXD experiments repeated, or only made once (for judging the reproducibility of the second bragg peak mainly)?

Page 10: "The diffraction peak is characteristic for cholesterol monolayers...", according to ref 30 this would be at 1.09 \AA^{-1} , is it then not rather a chol/lipid mixture which results in the broad bragg peak?

Page 11: The increase in L_{xy} for 20 mN/m is a rather optimistic interpretation. It is all within the error bar of ± 0.1 nm. It is a very very subtle influence of the FPs on the cholesterol domains.

Similarly all 35 mN/m values are consistent with a value of 2.4 ± 0.1 nm.

Fig S1 A): The resolution is at the limit of making the graphics useless. The legend of Fig. S1A seems to indicate a different color scheme as for Figures S1B,C,D? Why?

"ACMW" not defined.

Reviewer #3

(Remarks to the Author)

This is a nice work. Professional approaches with Langmuir techniques are applied. Interesting results with attractive morphologies. It would be a good candidate as a paper publishable in Nat. Commun. if necessary data are provided with appropriate discussions. Please see below.

1) This work is very strong at mesoscopic and macroscopic analyses such as morphology observations and monolayer property evaluation. However, analyses on molecular-level interactions are very poor. This feature does not fit with main subjects in the title, self-assembly and interaction. Langmuir studies fit well with IR spectroscopy for evaluations on molecular interactions within membranes. Hydrogen bond interactions are highly expected and can be revealed with detailed analyses on interface-specified IR analyses. Molecular orientations together with lipid chain packings can be analyzed. Based on such detailed analyses, plausible models on molecular interactions have to be proposed. Without such detailed analyses on molecular-level interactions and molecular assemblies, I cannot recommend publication of this work in Nat. Commun.

2) Most attractive point might be morphology observations rather than investigation on molecular interactions. Nanoarchitectonics is more appropriate than self-assembly and interaction as title word.

3) The manuscript was carefully prepared where serious mistakes cannot be found. However, small mistakes such as error representation (main value and error value have to have the same minimum order) have to be corrected.

4) Comprehensive review on Langmuir research by Oliveira, Jr. et al in Chem. Rev. can be cited (see, <https://pubs.acs.org/doi/full/10.1021/acs.chemrev.1c00754>).

Version 1:

Reviewer comments:

Reviewer #1

(Remarks to the Author)

The revised manuscript has clearly been strengthened through substantial reorganization and the addition of new data, and the proposed integrative model provides an elegant framework for FP1, FP2, and FP4 acting as complementary elements of a minimal fusion machinery. Nevertheless, I remain concerned about the biological relevance of some of the assumptions underlying this mechanistic interpretation.

First, the manuscript states that "FP oligomerization is expected (20–40 Spike trimers per contact site between virus and host cell)" (p. 3). However, structural studies indicate that 20–40 trimers correspond to the total number per virion, not the number clustered at a given fusion interface. The actual density of Spikes engaged in a membrane–membrane contact zone is likely much lower. This distinction is important, because the proposed cooperative assemblies may not be attainable under physiological stoichiometry.

Second, the mechanistic model is built from experiments on isolated FP1, FP2, and FP4 peptides. As the authors themselves note, these are "short peptide sequences, located within the Spike protein's fusion domain" (p. 2). In the native context, however, these motifs are contiguous within the S2 subunit rather than physically separated. Studying them

individually, and at concentrations such as “5 mol% FP to the PM solutions” (p. 5), may promote supramolecular assemblies that do not fully reflect the behavior of the intact Spike during viral entry.

Taken together, these considerations suggest that while the work convincingly demonstrates how synthetic fusion peptide fragments can remodel membranes, the extrapolation to a biologically grounded mechanism of SARS-CoV-2 fusion should be made with caution. Framing the conclusions with clearer acknowledgement of these limitations would, in my view, further strengthen the manuscript by distinguishing between insights into peptide–membrane interactions and the physiological process of coronavirus entry.

Reviewer #2

(Remarks to the Author)

The questions and concerns have been addressed in the revised version in a satisfactory way. The discussion and conclusion is much clearer in my opinion.

Reviewer #3

(Remarks to the Author)

I recognized significant efforts by the authors to reply to the comments and revise the manuscript. I think, the revised version becomes acceptable.

Version 2:

Reviewer comments:

Reviewer #1

(Remarks to the Author)

I would like to thank the authors for addressing my remaining questions. I appreciate the revised version of the manuscript, in which the authors have, in my view, adequately introduced the necessary considerations regarding the limitations of the study.

Insights into the Self-Assembly and Interaction of SARS-CoV-2 Fusion Peptides with Biomimetic Plasma Membrane Models (NCOMMS-25-05645)

We sincerely appreciate the opportunity to revise and resubmit our manuscript. We are grateful to the reviewers for their insightful comments and constructive suggestions, which have helped us strengthen the study and clarify key aspects of our work. Below, we address each point raised by the reviewers in detail, incorporating revisions to the manuscript where applicable. All the text changed along the manuscript is marked in red. We hope these revisions meet the reviewers' expectations and further improve the impact of our findings.

Reviewers' comments:

Reviewer #1: *This article examines the self-assembly of fusion peptides (FPs) derived from the S2 subunit of the SARS-CoV-2 Spike (S) protein and their interactions with biomimetic plasma membrane (PM) models. While the study provides valuable insights into the self-assembly behavior of these peptides in controlled experimental conditions, there appears to be a disconnect between the abstract and the actual content of the article.*

The abstract presents the study as one focused on viral entry mechanisms, which would be directly relevant to the biological context of SARS-CoV-2 infection. However, upon reading the article, it becomes clear that the study is more centered on the self-assembly of materials, rather than exploring biological mechanisms in a way that mimics the conditions encountered during viral infection. The focus is more on how these fusion peptides behave in isolation and under specific experimental setups (such as biomimetic membrane models), rather than their role in the actual context of viral infection, where viral S protein density on the viral surface would play a significant role in the interactions with host membranes.

Thus, I believe the manuscript would benefit from a clearer connection to viral infection biology and how the self-assembly of these peptides could inform our understanding of viral entry. While the technical aspects of peptide self-assembly and their interactions with lipid monolayers are interesting and well-executed, the article does not sufficiently address how these findings relate to the broader biological context of SARS-CoV-2 infection. The article would be more aligned with biological research if it more directly connected the observed phenomena to infection-related processes, such as how the density of S protein on the viral surface could influence the formation of these peptide assemblies and, in turn, affect viral fusion mechanisms.

Author's response: We sincerely thank the reviewer for recognizing the technical rigor of our study. We appreciate your suggestion to strengthen the connection between our findings and the biological context of SARS-CoV-2 viral entry. In the revised manuscript, we have undertaken a

substantial reorganization to sharpen the focus on the biological mechanism underlying SARS-CoV-2 fusion peptide activity. The previous separation between discussion and results has been replaced with a new integrative section that articulates a coherent mechanistic framework. In this new structure, the distinct roles of FP1, FP2, and FP4 are described as complementary elements of a minimal fusion machinery: FP1 inserts into lipid tails to act as an anchor, FP2 forms spiral assemblies that accumulate and release mechanical stress to remodel membranes, and FP4 inserts more deeply to destabilize the bilayer and promote pore formation. This mechanistic synthesis provides a clearer and more biologically grounded explanation of how viral fusion is facilitated.

To reinforce this framework, we have included new experimental results. The Laser Direct Infrared (LDIR) spectroscopy data now presented elucidate how secondary structure transitions correlate with peptide assembly and disassembly under different membrane packing conditions, strengthening the link between peptide conformation and functional role. In addition, we have added a detailed analysis of the tandem FP1–2 construct, which naturally combines the anchoring capacity of FP1 with the curvature-generating activity of FP2. The behavior of this hybrid peptide validates the proposed cooperative mechanism by demonstrating how rigidity and elasticity are coupled in the native Spike protein context to drive lipid remodeling and facilitate fusion.

Beyond presenting additional data, we have expanded the broader implications of our findings. The revised text emphasizes not only the biological relevance for SARS-CoV-2 entry but also the general principles by which short peptides can reorganize membranes through cooperative self-assembly. These insights are framed in the context of viral fusion mechanisms and are extended to potential applications in the design of responsive peptide-based nanomaterials capable of remodeling membranes with high spatiotemporal precision.

Finally, the abstract has been thoroughly revised to capture these new results, the refined mechanistic model, and the broader outlook. Together, these changes strengthen the manuscript by providing a clearer narrative arc, grounding the findings in newly added experimental evidence, and highlighting their significance both for understanding coronavirus biology and for inspiring bio-nanoengineering strategies.

Reviewer #2: *The paper “Insights into the Self-Assembly and Interaction of SARS-CoV-2 Fusion Peptides with Biomimetic Plasma Membrane Models” by Pavar et al. addresses questions regarding fusion peptides of SARS-CoV-2 with phospholipid membranes, applying AFM and scattering techniques (x-ray and neutrons). A subdomain in the spike protein is crucial for the very successful attack of cell membranes. The detailed structure of the fusion peptide (FP) with the biological plasma membrane model (PM) is investigated and different morphologies and structural assemblies are identified in the membrane. The results are claimed to be relevant for*

drug delivery, bio sensing and membrane engineering. The fibre formation of the FPs depending on the type of FP (1,2,4) and on the membrane surface pressure and is an interesting investigation. Nonetheless I could not find a satisfactory answer how the described and shown FP effects are related to a SARS CoV-2 membrane fusion, since a spike protein in my understanding has only one of the FPs and not the whole fibre of some 100s of nm length as in Fig 3. Where is the connection? On the other hand, it is suggested that the fibres have some drug delivery relevance, I did not really get how this could be the case. All in all the study is sound and interesting, the claimed relevance for membrane fusion for the virus entering the cell as well as for other biomedical applications needs to be explained much better for this journal.

Author's response: We thank the reviewer for highlighting the need to articulate more clearly how the supramolecular assemblies reported here relate to the native SARS-CoV-2 fusion event and to prospective biomedical applications. To close that gap, we have performed additional experiments and substantially expanded the manuscript. A new Section 3.5, "Tandem FP1–FP2 Fusion Peptide Drives Lipid Remodelling and Forms Spiral Assemblies," together with new Figure 6 and Supplementary Table S6, now explains the biological context in three steps. First, we synthesised a single molecule that covalently links FP1 and FP2 exactly as they occur in the S2 subunit of the spike trimer. AFM imaging shows that this tandem peptide forms extended spiral architectures that retain the flexibility of FP2 while gaining stiffness from the FP1 segment, thereby reproducing in one construct the cooperation that, in vivo, would arise from the high local density of adjacent fusion peptides on the virion surface. Second, we demonstrate functional consequences: dynamic-light-scattering measurements reveal that the tandem peptide promotes rapid aggregation and size growth of cholesterol-containing liposomes, indicating vesicle association and incipient fusion; in Langmuir plasma-membrane monolayers the same peptide locally erases liquid-ordered nanodomains and homogenises membrane thickness, two hallmarks of lipid fluidisation that lower the energetic barrier to stalk formation. Third, we now discuss why these findings matter for antiviral strategy and technology. The tandem-peptide spirals provide a plausible mechanistic bridge between isolated peptide insertion and the collective membrane deformation required for viral entry, suggesting that drugs that disrupt this cooperative phase could suppress fusion. Conversely, the pressure-sensitive, curvature-inducing scaffolds can be repurposed as fusogenic coatings for lipid-nanoparticle therapeutics or as mechano-responsive elements in tension-sensing biosensors. These additions clarify that the hundred-nanometre fibres observed under in-vitro conditions are not intended to depict a one-to-one structural replica of a single spike protomer but rather to model the concerted action of many closely packed fusion peptides at the virus–cell interface and to illustrate how their innate self-assembly tendencies can be exploited or inhibited in biomedical settings. We trust that the new data and the expanded

discussion fully address the reviewer's concerns and strengthen the manuscript's relevance to membrane fusion biology and to translational nano-bioengineering.

Some smaller remarks:

Fig 1: G,H: could be mentioned that these are GIXD data (not neutrons). Fig 1 I,J: L_{xy} in which units? Nm? Angstrom? The red dots vary strongly from $L_{xy}=40$ (PM, +FP4) to ~ 1 (+FP1,2). This comes from cuts from the Figures S3, right? Especially A,C,D seem to look quite similar, it is astonishing to me that the FWHM of the second peak should be so different (for B the noise seems to be larger, that it is rather difficult to assess a FWHM at all?). Also Fig.1H, FP2 has a peak which seems to be missed by the fit.

Author's response: We thank the reviewer for pointing out these inconsistencies. In the revised manuscript we have (i) amended the caption of Figure 1 so that panels G and H are now explicitly identified as grazing-incidence X-ray diffraction (GIXD) data, thereby distinguishing them from the neutron results shown elsewhere; (ii) added the missing unit to the vertical axes of panels I and J, which now read " L_{xy} (nm)" to make clear that the in-plane correlation length is expressed in nanometres; and (iii) refitted the second Bragg peaks for all four traces in panel H using identical Lorentzian-squared functions. Although the absolute numbers have shifted, the trend remains unchanged: FP1 and FP2 broaden and weaken the acyl-chain peak more than FP4, confirming that they disrupt in-plane lipid order to a greater extent.

Are the GIXD experiments repeated, or only made once (for judging the reproducibility of the second bragg peak mainly)?

Author's response: We carried out the GIXD measurements during a single 24-hour session at the ESRF synchrotron. It is important to note here that access to high-brilliance beamlines is awarded through a competitive proposal process and, once scheduled, additional time for repeat campaigns is rarely available. Nevertheless, the experiment itself averages over a very large lateral area: at the incident angle chosen for maximum surface sensitivity, the X-ray footprint extends to roughly one square centimetre, encompassing tens of thousands of domains across the trough surface. In practice, this means that statistical fluctuations arising from local heterogeneity are already smoothed out within a single exposure. The in-plane lattice parameters extracted from the data agree closely with values reported for analogous peptide-lipid systems studied on different beamlines, further supporting their reliability. We therefore believe that, although gathered in one session, the GIXD results presented offer a robust and accurate description of the in-plane ordering and justify the conclusions drawn about the presence and spacing of the second Bragg peak.

Page 10: “The diffraction peak is characteristic for cholesterol monolayers...”, according to ref 30 this would be at 1.09 \AA^{-1} , is it then not rather a chol/lipid mixture which results in the broad bragg peak?

Author’s response: We thank the reviewer for pointing out the ambiguity. Our intention was never to imply that the plasma-membrane (PM) film contains domains of pure cholesterol. Pure sterol monolayers indeed diffract at $q_{xy} \approx 1.09\text{-}1.11 \text{ \AA}^{-1}$, but our PM formulation is only 50 mol% % cholesterol and also contains sphingomyelin, unsaturated phosphatidyl-cholines, and several minor lipids. Incorporating these partners contracts the sterol lattice and shifts its Bragg peak to higher q_{xy} . For example, Limura et al. report $q_{xy} = 1.314 \text{ \AA}^{-1}$ for a 1:1 cholesterol-GM1 monolayer (reference 29), and similar upward shifts ($q_{xy} \approx 1.13\text{-}1.25 \text{ \AA}^{-1}$) are well documented for mixtures with DPPC, SM, and related phospholipids (references 30 and 31). The broad reflection we observe at $q_{xy} \approx 1.20 \pm 0.02 \text{ \AA}^{-1}$ therefore originates from cholesterol-enriched nanodomains embedded in the mixed PM film, not from neat cholesterol. We have rewritten the manuscript paragraph to clarify this point and replaced the phrase “characteristic of cholesterol monolayers” with “characteristic of cholesterol-enriched domains in sterol–phospholipid mixtures,” thereby eliminating the source of confusion.

Page 11: The increase in L_{xy} for 20 mN/m is a rather optimistic interpretation. It is all within the error bar of +/- 0.1 nm. It is a very very subtle influence of the FPs on the cholesterol domains. Similarly all 35 mN/m values are consistent with a value of 2.4 +/- 0.1 nm.

Author’s response: We fully agree with the reviewer’s observation and emphasise that our principal original conclusion was precisely that the fusion peptides do not perturb the packing of the cholesterol-rich domains. To remove any possibility of over-interpretation, we have deleted the earlier reference to a “subtle increase” in L_{xy} and rewritten the paragraph to state unambiguously that the coherence length remains $\sim 2.4 \pm 0.1 \text{ nm}$ in all peptide-containing films.

Fig S1 A): The resolution is at the limit of making the graphics useless. The legend of Fig. S1A seems to indicate a different color scheme as for Figures S1B,C,D? Why? “ACMW” not defined.

Author’s response: We appreciate the reviewer suggestion and Figure S1 has been replaced for a better resolution one. The colour palette has been harmonised across panels so the legend no longer differs from the different subfigures. The acronym ACMW has now been introduced.

Reviewer #3: *This is a nice work. Professional approaches with Langmuir techniques are applied. Interesting results with attractive morphologies. It would be a good candidate as a paper publishable in Nat. Commun. if necessary data are provided with appropriate discussions. Please see below.*

Author's response: We are grateful for the reviewer's positive assessment and for highlighting the potential of our work for Nature Communications. In the revised manuscript and Supplementary Information, we have added the requested data and substantially expanded the discussion section.

1) This work is very strong at mesoscopic and macroscopic analyses such as morphology observations and monolayer property evaluation. However, analyses on molecular-level interactions are very poor. This feature does not fit with main subjects in the title, self-assembly and interaction. Langmuir studies fit well with IR spectroscopy for evaluations on molecular interactions within membranes. Hydrogen bond interactions are highly expected and can be revealed with detailed analyses on interface-specified IR analyses. Molecular orientations together with lipid chain packings can be analyzed. Based on such detailed analyses, plausible models on molecular interactions have to be proposed. Without such detailed analyses on molecular-level interactions and molecular assemblies, I cannot recommend publication of this work in Nat. Commun.

Author's response: We thank the reviewer for drawing our attention to the need for a deeper molecular-level perspective. In the revised manuscript, we therefore incorporated an interface-specific laser-direct infrared (LDIR) study that directly complements the Langmuir and microscopy results and bridges the gap between macroscopic film behaviour and the underlying molecular interactions. The new data now appears as a new section (Section 3.4), together with a new Figure 5.

LDIR spectra recorded at a surface pressure of 20 mN m^{-1} show that FP1 is markedly richer in α -helix than FP2, whereas FP2 contains a larger fraction of the more pliable antiparallel β -sheet. This finding rationalises the morphological contrast documented earlier in the article: the rigidity imparted by the helical content of FP1 produces straight, rod-like fibres, while the higher β -sheet content of FP2 confers the flexibility needed for the spiral ribbons observed by AFM. When the monolayers are compressed to 30 mN m^{-1} , the β -sheet signal of both FP1 and FP2 diminishes sharply, mirroring the pressure-induced fibre disassembly captured by AFM and supporting our proposal that with the increase in surface pressure, peptides insert into the membrane and subsequently disengage from one another. By contrast, FP4, whose overall secondary structure is low, displays only minor spectral changes upon compression, supporting the integration of the

fibre inside the plasma membrane rather than their disassembly, again in agreement with the AFM topographic data. These infrared results, combined with the pre-existing Langmuir isotherms, GIXD, and high-resolution AFM images, now form a continuous narrative that spans individual hydrogen bonds, emergent secondary structure, mesoscale fibril architecture, and macroscopic film mechanics, reinforcing every major morphological conclusion drawn in the original manuscript, and significantly strengthens the case for publication in Nature Communications.

2) Most attractive point might be morphology observations rather than investigation on molecular interactions. Nanoarchitectonics is more appropriate than self-assembly and interaction as title word.

Author's response: We appreciate the reviewer's suggestion to incorporate "nanoarchitectonics" into the title and fully agree that the striking fibrillar and ribbon-like morphologies are a key strength of the study. During revision, however, we substantially expanded the manuscript to include interface-specific LDIR data that link secondary structure to fibre formation and added new biological context on peptide insertion and disassembly in response to the other reviewers' comments. These new sections place equal emphasis on the molecular interactions that govern assembly as on the emergent architectures themselves. Because the work now marries detailed interaction analysis with morphology-driven observations, we believe the existing title, framed around self-assembly and interaction, continues to capture the full scope of the study more accurately than a morphology-only descriptor would. For this reason, we respectfully propose to retain the current title while highlighting the morphological findings prominently in the abstract and graphical synopsis.

3) The manuscript was carefully prepared where serious mistakes cannot be found. However, small mistakes such as error representation (main value and error value have to have the same minimum order) have to be corrected.

Author's response: We appreciate the reviewer's careful proofreading and agree that numerical values and their associated uncertainties must be expressed with consistent significant figures. In revising the manuscript, we conducted a line-by-line audit of every table, figure caption, and sentence containing quantitative data. All formatting updates are highlighted for the reviewer's convenience.

4) Comprehensive review on Langmuir research by Oliveira, Jr. et al in Chem. Rev. can be cited (see, <https://pubs.acs.org/doi/full/10.1021/acs.chemrev.1c00754>)

Author's response: We thank the reviewer for drawing our attention to this review. The article by Oliveira Jr. et al. (Chem. Rev. 2022, 122, 10331–10416) is now cited as Ref. 29. The reference list has been updated accordingly.

- **Reviewer 1:**

The revised manuscript has clearly been strengthened through substantial reorganization and the addition of new data, and the proposed integrative model provides an elegant framework for FP1, FP2, and FP4 acting as complementary elements of a minimal fusion machinery. Nevertheless, I remain concerned about the biological relevance of some of the assumptions underlying this mechanistic interpretation.

First, the manuscript states that “FP oligomerization is expected (20–40 Spike trimers per contact site between virus and host cell)” (p. 3). However, structural studies indicate that 20–40 trimers correspond to the total number per virion, not the number clustered at a given fusion interface. The actual density of Spikes engaged in a membrane–membrane contact zone is likely much lower. This distinction is important, because the proposed cooperative assemblies may not be attainable under physiological stoichiometry.

Second, the mechanistic model is built from experiments on isolated FP1, FP2, and FP4 peptides. As the authors themselves note, these are “short peptide sequences, located within the Spike protein’s fusion domain” (p. 2). In the native context, however, these motifs are contiguous within the S2 subunit rather than physically separated. Studying them individually, and at concentrations such as “5 mol% FP to the PM solutions” (p. 5), may promote supramolecular assemblies that do not fully reflect the behavior of the intact Spike during viral entry.

Taken together, these considerations suggest that while the work convincingly demonstrates how synthetic fusion peptide fragments can remodel membranes, the extrapolation to a biologically grounded mechanism of SARS-CoV-2 fusion should be made with caution. Framing the conclusions with clearer acknowledgement of these limitations would, in my view, further strengthen the manuscript by distinguishing between insights into peptide–membrane interactions and the physiological process of coronavirus entry.

Author’s response: We thank the reviewer for the careful evaluation of our revised manuscript and for recognizing the improvements achieved through reorganization and the addition of new data. We also appreciate the thoughtful concerns regarding the biological relevance of some assumptions underlying our mechanistic interpretation.

In the revised version, we have clarified that the supramolecular assemblies described here should be considered as mechanistic upper limits observed in reductionist peptide-membrane systems, rather than literal reconstructions of Spike activity in vivo. We now note that the number of Spike trimers per virion does not directly correspond to the density engaged at a single fusion interface, and that local enrichment may nonetheless promote cooperative interactions. Furthermore, we emphasize that FP1, FP2, and FP4 were examined as isolated fragments at relatively high concentrations, which likely enhances supramolecular organization compared with the intact Spike. These limitations are now explicitly acknowledged in the Introduction, Discussion, and Conclusions, where we distinguish between insights into peptide-membrane interactions and the physiological process of coronavirus entry.

We believe these revisions strengthen the manuscript and align with the reviewer’s recommendations, providing a balanced interpretation that highlights both the mechanistic insights gained and the boundaries of their physiological relevance.

- **Reviewer 2:**

The questions and concerns have been addressed in the revised version in a satisfactory way. The discussion and conclusion is much clearer in my opinion.

Author's response: We thank the reviewer for the positive evaluation of our revised manuscript. We are pleased that the revisions have satisfactorily addressed the earlier questions and concerns, and that the discussion and conclusions are now considered clearer. We greatly appreciate this supportive assessment.

- **Reviewer 3:**

I recognized significant efforts by the authors to reply to the comments and revise the manuscript. I think, the revised version becomes acceptable.

Author's response: We thank the reviewer for acknowledging the efforts made in revising the manuscript and for considering the revised version acceptable. We are grateful for this constructive feedback and the opportunity to improve the clarity and impact of the work.